# Effect of Aging on the Trunk and Lower Limb Kinematics during Gait on a Compliant Surface in Healthy Individuals

Keita Honda [1,2,*], Yusuke Sekiguchi [2] and Shin-Ichi Izumi [2,3]

1 Department of Rehabilitation, Kumamoto Health Science University, 325 Izumi-machi, Kita-ku, Kumamoto 861-5598, Japan

2 Department of Physical Medicine and Rehabilitation, Tohoku University Graduate School of Medicine, 2-1 Seiryo-machi, Aoba-ku, Sendai 980-8575, Japan

3 Graduate School of Biomedical Engineering, Tohoku University, 2-1 Seiryo-machi, Aoba-ku, Sendai 980-8575, Japan

* Correspondence: honda.kt@kumamoto-hsu.ac.jp; Tel.: +81-96-275-2162

**Abstract:** Older adults have a smaller effective living space and reduced physical activity. Although walking ability in various living spaces is necessary to maintain a healthy life and a high level of physical activity, it is unclear how older adults adapt to compliant surfaces when walking. The purpose of this study was to determine the differences in the trunk and lower limb kinematics while walking on a level versus compliant surface, and the effect of aging on these kinematic changes. Twenty-two healthy individuals (aged from 20–80 years) were asked to walk along a 7-m walkway at a comfortable speed on a level and compliant surface. Gait kinematics were measured using a three-dimensional camera-based motion analysis system. We found that knee and hip flexion and ankle plantarflexion angles in the early stance phase and thoracic flexion angle throughout the gait cycle were significantly increased when walking on a compliant surface versus a level surface. The change in the thoracic flexion angle, ankle plantarflexion angle, and cadence between level and compliant surfaces was significantly correlated with age. Therefore, older adults use increased thoracic flexion and ankle plantarflexion angles along with a higher cadence to navigate compliant surfaces.

**Keywords:** gait; kinematic; adaptation; aging; uneven ground

## 1. Introduction

Maintaining high physical activity levels is important for a healthy lifestyle. Daily physical activity is assessed by the number of steps per day and inactivity is associated with risk of death [1] and stroke [2], stroke-related severity [3], and prefrail status [4]. Notably, the intensity of physical activity is associated with greater living space in the life-space assessment, and walking ability is essential to sustain this physical activity in large living spaces [5]. A previous study reported that older adults living in large spaces have superior physical functioning levels, which include better walking speed, lower extremity muscle strength, walking endurance, motor skills in walking, balance ability, and energy cost of walking [6]. Since indoor walking is not sufficient for adequate physical activity, walking ability in different living spaces is necessary to maintain a healthy life and a high level of physical activity.

Current biomechanical research is increasingly focused on walking on level surfaces as well as simulated uneven terrain in the laboratory and on walkways in real life [7–11]. It is known that walking on compliant surfaces, such as sand or a soft gym mat, consumes approximately 2–3 times more energy expenditure than walking on level surfaces [7,8]. Furthermore, the total mechanical work on a compliant surface is approximately 1.6–2.5 times greater than that on a level surface [8]. Adaptive changes in the kinematics of the trunk and lower limbs while walking on a compliant surface may be one of the reasons for the increased mechanical work and energy expenditure. Additionally, it has been observed

that increasing the hip and knee flexion angles during the early stance and swing phases lowers the vertical position of the center of mass (CoM) to increase stability while walking on compliant surfaces [9,10]. However, these findings are based on healthy young adults. Although Barbara, Freitas, Bagesteiro, Perracini, and Alouche [11] found that older adults aged 80 and older had greater displacement of the pelvic segment during gait on compliant surfaces than younger older adults aged 65–75 years, this study did not include adults less than 65 years old. It is important to examine how older adults adapt to compliant surfaces during walking, since they tend to have smaller living spaces, which is the effective area they traverse during their daily life and during reduced physical activity. Although effects of hard, uneven terrain on gait for older adults has been examined [12,13], it is unclear if the effects of aging impact gait characteristics on compliant surfaces. Understanding the effects of aging on gait characteristics on compliant surfaces may help identify the gait parameters that need to be addressed for older adults to navigate different environments.

This study aimed to determine the differences in the kinematic characteristics of the lower limbs and trunk while walking on a level versus compliant surface, and the effect of aging on these kinematic changes. We hypothesized that the increase in knee and hip flexion angles in the early stance and swing phase observed in healthy young adults when walking on a compliant surface would be more pronounced in older adults to increase stability during gait.

## 2. Materials and Methods

### 2.1. Participants

Twenty-two healthy individuals satisfying the following inclusion criteria participated in this study: (1) aged 20–80 years; (2) ability to walk on level and compliant surfaces without assistance; (3) a walk score of 7 on the Functional Independence Measure; (4) Functional Ambulation Category: 5; (5) able to follow verbal commands (Table 1). Participants were excluded in the case of (1) abnormal circulatory and respiratory status (i.e., shortness of breath when walking on level ground); (2) a history of neurological and orthopedic problems that interfere with gait; (3) abnormal mental status.

**Table 1.** Characteristics of study participants (n = 22).

| Characteristic | |
|---|---|
| Gender (males, females) [a] | 12, 10 |
| Age (years) [b] | 43.5 (21.5) |
| 20–29 years [a] | 11 |
| 30–39 years [a] | 1 |
| 40–49 years [a] | 2 |
| 50–59 years [a] | 2 |
| 60–69 years [a] | 1 |
| 70–79 years [a] | 5 |
| Height (cm) [b] | 165.7 (7.5) |
| Weight (kg) [b] | 59.8 (8.1) |

[a] number of participants; [b] mean (standard deviation).

### 2.2. Experimental Procedure

The participants were instructed to walk along a 7-m walkway at a comfortable speed on the level and compliant surfaces. The participants did not use any assistive device, such as a cane or orthosis. For the level surface, the typical laboratory tiled walkway was used; for the compliant surface, we set up a 7-m walkway using an AIREX mat (OLYMPIA AMG-200G; SAKAI Medical Co., Ltd., Tokyo, Japan) (Figure 1). The AIREX mat is made of a material containing air bubbles and thus has elasticity. The AIREX mat is used to induce instability during standing tasks in rehabilitation and has a smooth surface structure and soft material properties [14]. The order of surface used was randomly assigned to each participant. A rest period was between two surfaces sessions. After a 5 min practice on each floor to familiarize the participants, we collected data for five trials on each surface

condition for each participant. One to three gait cycles were collected for each trial. The collection time for each surface condition was 3 to 5 min.

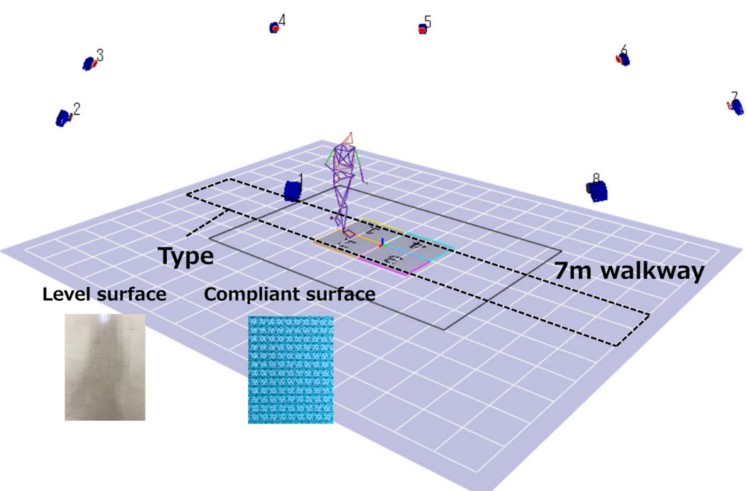

**Figure 1.** Participants walked on the level surface and compliant surface.

Forty reflective markers were attached to thirteen body segments (Figure 2), consisting of the head, thorax, pelvis, thighs, shanks, feet, upper arms, and forearms, based on the previous reports [15,16]. An 8-camera motion analysis system (MAC 3D; Motion Analysis Corp., Santa Rosa, CA, USA) and four 90 × 60 cm force plates (Anima Corp., Tokyo, Japan) were used to measure data regarding the three-dimensional (3D) coordinates of these markers and the ground reaction force (GRF). The motion analysis system and force plates were synchronized for data collection (sampling frequency = 120 Hz and 1200 Hz, respectively).

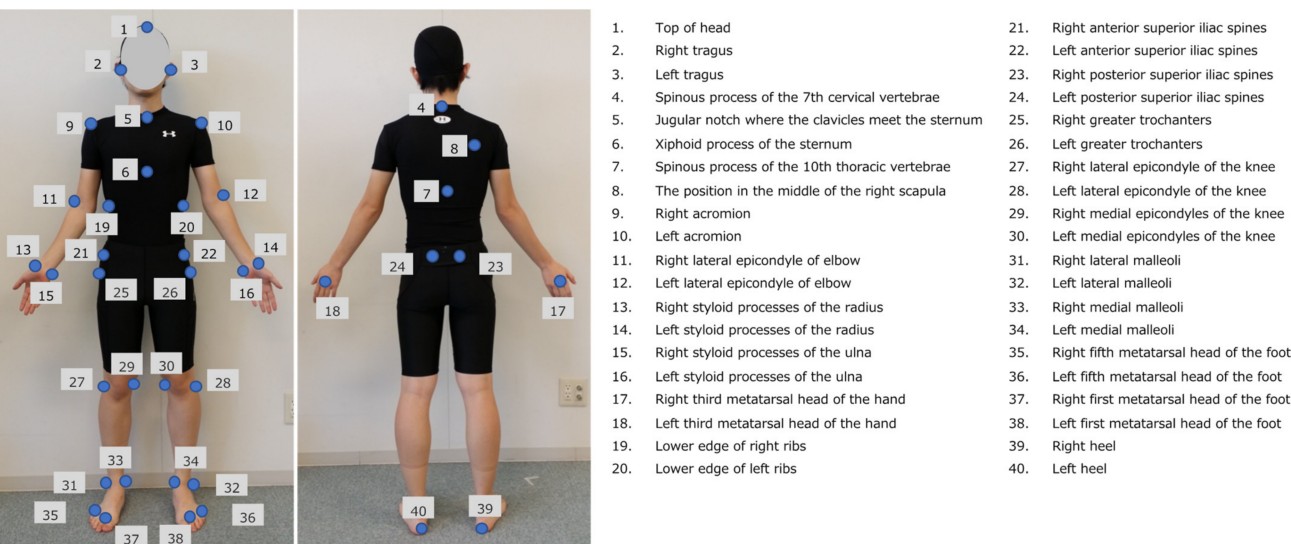

1. Top of head
2. Right tragus
3. Left tragus
4. Spinous process of the 7th cervical vertebrae
5. Jugular notch where the clavicles meet the sternum
6. Xiphoid process of the sternum
7. Spinous process of the 10th thoracic vertebrae
8. The position in the middle of the right scapula
9. Right acromion
10. Left acromion
11. Right lateral epicondyle of elbow
12. Left lateral epicondyle of elbow
13. Right styloid processes of the radius
14. Left styloid processes of the radius
15. Right styloid processes of the ulna
16. Left styloid processes of the ulna
17. Right third metatarsal head of the hand
18. Left third metatarsal head of the hand
19. Lower edge of right ribs
20. Lower edge of left ribs
21. Right anterior superior iliac spines
22. Left anterior superior iliac spines
23. Right posterior superior iliac spines
24. Left posterior superior iliac spines
25. Right greater trochanters
26. Left greater trochanters
27. Right lateral epicondyle of the knee
28. Left lateral epicondyle of the knee
29. Right medial epicondyles of the knee
30. Left medial epicondyles of the knee
31. Right lateral malleoli
32. Left lateral malleoli
33. Right medial malleoli
34. Left medial malleoli
35. Right fifth metatarsal head of the foot
36. Left fifth metatarsal head of the foot
37. Right first metatarsal head of the foot
38. Left first metatarsal head of the foot
39. Right heel
40. Left heel

**Figure 2.** Marker locations for motion analysis.

### 2.3. Data Analysis

The 3D marker coordinates and GRF data were smoothed using a fourth-order Butterworth low-pass filter (cutoff frequency = 6 Hz and 80 Hz, respectively). The cutoff frequency of smoothing for GRF data was determined according to the previous study [17]. Gait cycle events (i.e., initial contact (IC) and toe-off (TO)) were detected using vertical GRF data; the threshold value was set at 20 N. We analyzed the gait cycle for the right

foot, i.e., from right IC to subsequent ipsilateral IC. The 13-segment link model for the markers was constructed based on the segmental coordinate system for the head, thorax, pelvis, thighs, shanks, feet, upper arms, and forearms; ankle, knee, and hip joint angles were calculated using this joint coordinate system [16]. To evaluate the characteristics of lower limb movements during walking, we calculated the ankle, knee, and hip joint kinematic variables proposed by Benedetti et al. [18]. Furthermore, the foot angle with respect to the floor in the sagittal plane was calculated to examine the foot angle at IC. In addition, the thoracic and pelvic angles in the sagittal plane were defined as the angle of the longitudinal axis of the thorax or pelvis with respect to the global coordinate system. The mean and peak-to-peak values of the thoracic and pelvic angles in the sagittal planes were calculated throughout one gait cycle to evaluate the characteristics of trunk movements during walking. A custom program in MATLAB R2019b (The Math Works, Inc., Natick, MA, USA) was used to compute these parameters.

### 2.4. Statistical Analysis

The mean value of five gait cycles for each participant was used for the statistical analysis. The normality of distribution for all parameters was confirmed using the Shapiro–Wilk test. Since some variables (foot angle at IC on the level surface, peak-to-peak value of hip joint angle throughout the gait cycle on the level surface, peak-to-peak value of thoracic angle throughout the gait cycle on the level surface, difference in stride length between conditions, difference in knee flexion angle at IC between conditions, difference in maximum ankle plantarflexion angle in 1st half of the stance phase between conditions, and difference in maximum ankle dorsiflexion angle in the swing phase between conditions) were not normally distributed and the sample size was small, nonparametric statistics were used to test all gait variables. Gait variables were compared between level and compliant surfaces using the Wilcoxon signed-rank test. The effect size (r) was calculated as $r = Z/\sqrt{N}$, where "Z" is computed using the Wilcoxon signed-rank test and "*N*" is the number of participants. Small, medium, and large effect sizes (r) were determined as 0.10, 0.30, and 0.50, respectively [19]. Spearman's rank correlation coefficients were used to determine whether the difference in gait variables between level and compliant surfaces ($variable_{compliant} - variable_{level}$) was related to age. A statistical significance level of $p < 0.05$ was used. Statistical analyses were performed with the Statistical Package for the Social Sciences (version 24.0, IBM Corporation, Armonk, NY, USA).

An a priori power analysis was performed using G*Power 3.1.9.4 (Heinrich-Heine University, Düsseldorf, Germany) based on the knee flexion angle in the early stance phase; this variable often shows kinematic changes according to the surface conditions [9,10]. The power analysis identified a minimum sample size required to obtain sufficient statistical power $(1-\beta = 0.80)$ at $\alpha = 0.05$ [20,21]. Based on the pilot data for the knee flexion angle in the early stance phase collected from 10 subjects, an effect size (d) was assumed as 0.947; accordingly, a minimum total sample of 22 participants was recommended.

## 3. Results

### 3.1. Spatiotemporal Gait Parameters While Walking on Level and Compliant Surfaces

Step width was significantly larger on the compliant surface than that on the level surface ($p < 0.001$); there were no significant differences in other spatiotemporal parameters (Table 2).

### 3.2. Kinematic Variables during Gait on the Level and Compliant Surface

The average of the lower limb joint, thoracic, and pelvic angles are shown in the sagittal plane during the gait cycle on the level surface and compliant surface (Figure 3). The thoracic flexion angle was larger when walking on the compliant surface as compared to walking on the level surface. In addition, the ankle, knee, and hip joint angles differed between the conditions from the swing phase to the early stance phase.

**Table 2.** A comparison of spatiotemporal gait parameters on different walking surfaces.

| | Floor Surface | | Statistical Value | |
|---|---|---|---|---|
| | **Level** | **Compliant** | ***p*-Value** | **Effect Size, r** |
| Walking speed (m/s) | 1.27 (0.19) | 1.30 (0.23) | 0.062 | −0.398 |
| Step length (m) | | | | |
| Left side | 0.62 (0.06) | 0.63 (0.07) | 0.372 | −0.19 |
| Right side | 0.63 (0.07) | 0.66 (0.06) | 0.062 | −0.398 |
| Stride length (m) | 1.29 (0.17) | 1.35 (0.15) | 0.101 | −0.35 |
| Step width (m) | 0.14 (0.03) | 0.16 (0.05) | <0.001 | −0.765 |
| Cadence (steps/min) | 111.3 (9.1) | 111.6 (10.0) | 0.123 | −0.329 |

All floor surface values are median (interquartile range).

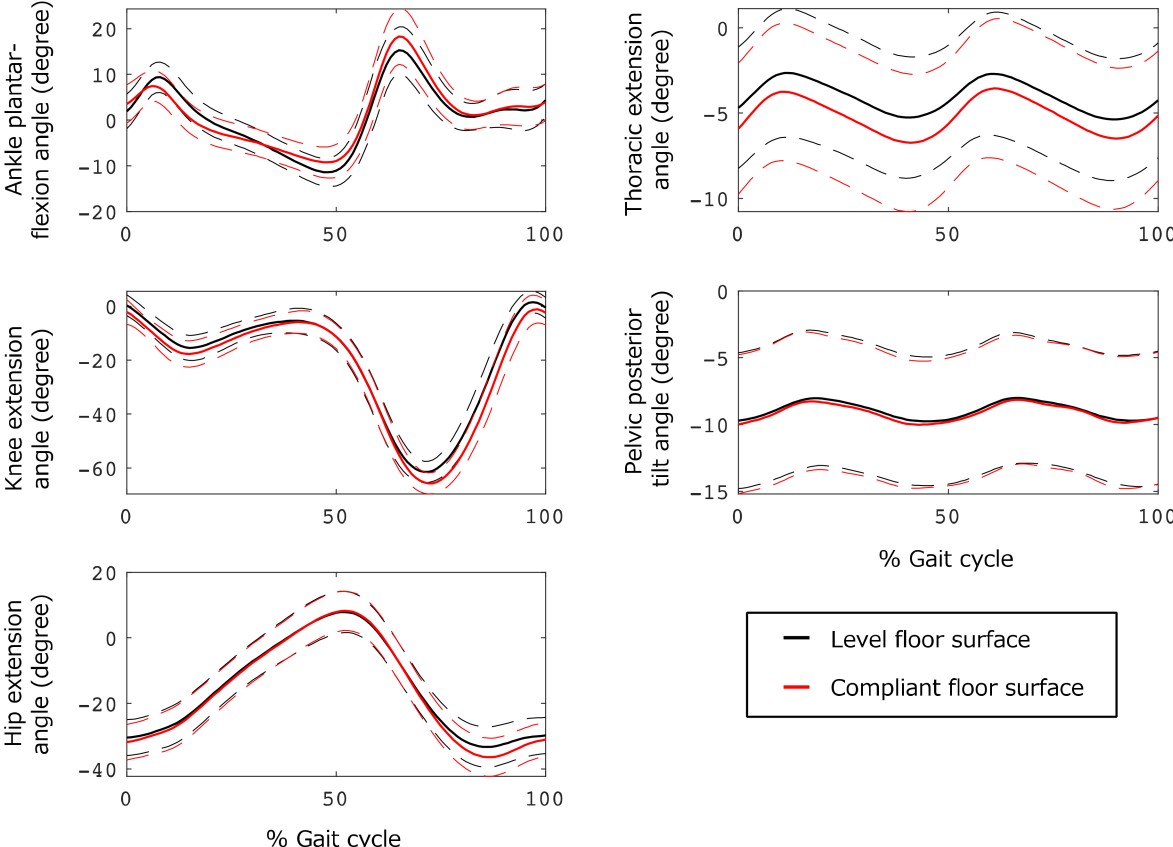

**Figure 3.** Average (solid line) and 1 SD (dashed line) of joint angle (left line) and segment angle (right line) during gait cycle on the level surface (black) and compliant surface (red).

A comparison of the lower limb kinematics between the level and compliant surface conditions is shown in Table 3. The hip flexion angle at IC ($p$ = 0.002) and maximum hip flexion angle during the stance ($p$ = 0.002) and swing ($p$ < 0.001) phases were significantly greater when walking on the compliant surface as compared to walking on the level surface, while there was no significant difference in the maximum hip extension angle ($p$ = 0.123). Similarly, the knee flexion angle at IC ($p$ < 0.001) and maximum knee flexion angle in the first half of the stance ($p$ < 0.001) and swing phases ($p$ < 0.001) were significantly greater on the compliant surface. In the ankle joint, although the ankle plantarflexion angle at IC ($p$ = 0.003) was significantly greater on the compliant surface, the maximum ankle plantarflexion angle in the first half of the stance phase ($p$ < 0.001) and the maximum ankle dorsiflexion angles in the second half of stance phase ($p$ < 0.001) were significantly smaller in the compliant surface condition than in the level surface condition. The foot angle with respect to the floor in the sagittal plane was significantly smaller in the compliant surface condition than in the level surface condition ($p$ = 0.006).

**Table 3.** Hip, knee, and ankle joint angles during gait on the level and compliant surfaces.

| | Floor Surface | | Statistical Value | |
| --- | --- | --- | --- | --- |
| | Level | Compliant | *p*-Value | Effect Size, r |
| **Hip joint angle (degree) [positive value: extension]** | | | | |
| Flexion angle at initial contact | −30.5 (7.6) | −32.9 (7.3) | **0.002** | 0.67 |
| Maximum flexion angle in 1st half of the stance phase | −30.5 (7.8) | −33.0 (7.4) | **0.002** | 0.65 |
| Maximum extension angle in stance phase | 6.8 (6.8) | 6.8 (5.3) | 0.123 | −0.33 |
| Extension angle at toe-off | −0.5 (9.1) | −0.9 (5.2) | 0.263 | 0.24 |
| Maximum flexion angle in the swing phase | −34.6 (7.4) | −37.7 (6.8) | **<0.001** | 0.88 |
| Peak-to-peak value throughout the gait cycle | 41.0 (5.6) | 45.1 (4.1) | **<0.001** | −0.88 |
| **Knee joint angle (degree) [positive value: extension]** | | | | |
| Flexion angle at initial contact | 0.1 (4.8) | −1.8 (6.9) | **<0.001** | 0.85 |
| Maximum flexion angle in 1st half of the stance phase | −14.1 (3.4) | −17.3 (7.0) | **<0.001** | 0.83 |
| Maximum extension angle in 2nd half of stance phase | 0.3 (4.8) | −1.8 (4.6) | **<0.001** | 0.85 |
| Flexion angle at toe-off | −39.9 (5.2) | −42.7 (3.3) | **0.001** | 0.73 |
| Maximum flexion angle in the swing phase | −61.9 (5.9) | −65.4 (4.2) | **<0.001** | 0.88 |
| Peak-to-peak value throughout the gait cycle | 63.9 (5.6) | 66.2 (6.3) | **0.001** | −0.70 |
| **Ankle joint angle (degree) [positive value: plantarflexion]** | | | | |
| Dorsiflexion angle at initial contact | 1.6 (3.3) | 3.7 (4.8) | **0.003** | −0.64 |
| Maximum plantarflexion angle in 1st half of the stance phase | 9.8 (3.8) | 7.5 (3.7) | **<0.001** | 0.81 |
| Maximum dorsiflexion angle in 2nd half of stance phase | −11.6 (3.3) | −10.2 (4.6) | **<0.001** | −0.85 |
| Plantarflexion angle at toe-off | 10.7 (5.1) | 13.4 (9.5) | **<0.001** | −0.83 |
| Maximum dorsiflexion angle in the swing phase | −1.1 (4.9) | −0.3 (4.8) | **0.020** | −0.49 |
| Peak-to-peak value throughout the gait cycle | 28.4 (5.7) | 27.2 (7.8) | 0.485 | −0.15 |
| **Foot angle (degree) [positive value: toe higher than heel]** | | | | |
| Angle at initial contact | 18.5 (3.6) | 15.6 (6.3) | **0.006** | 0.59 |

All floor surface values are median (interquartile range). Bold *p*-values represent significant differences between conditions (*p* < 0.05).

Regarding the trunk kinematics, the thoracic flexion angle was significantly greater on the compliant surface than that on the level surface (*p* < 0.001, Table 4). Furthermore, the peak-to-peak value for thoracic flexion and extension on the compliant surface was significantly greater than that on the level surface (*p* = 0.003).

**Table 4.** Trunk angle during walking on the level and compliant surface.

| | Floor Surface | | Statistical Value | |
| --- | --- | --- | --- | --- |
| | Level | Compliant | *p*-Value | Effect Size, r |
| **Thoracic angle (degree) [positive value: extension]** | | | | |
| Mean value throughout the gait cycle | −4.8 (5.6) | −6.3 (6.9) | **<0.001** | 0.87 |
| Peak-to-peak value throughout the gait cycle | 3.7 (0.6) | 4.1 (0.9) | **0.003** | −0.63 |
| **Pelvic angle (degree) [positive value: posterior tilt]** | | | | |
| Mean value throughout the gait cycle | −10.1 (6.6) | −9.8 (7.3) | 0.306 | 0.22 |
| Peak-to-peak value throughout the gait cycle | 3.2 (1.0) | 3.3 (0.7) | 0.291 | −0.22 |

All floor surface values are median (interquartile range). Bold *p*-values represent significant differences between conditions (*p* < 0.05).

*3.3. Correlation between Age and Floor Surface-Related Changes in Gait Variables*

The results of the correlation analysis between age and floor surface-related changes in gait variables are shown in Table 5. Significant positive correlations were observed between the participants' age and step width (r = 0.60, *p* = 0.003), cadence (r = 0.60, *p* = 0.003), ankle dorsiflexion angle at IC (r = 0.59, *p* = 0.004), and maximum ankle dorsiflexion angle in the swing phase (r = 0.61, *p* = 0.002). On the other hand, stride length (r = −0.44, *p* = 0.042), peak-to-peak value of ankle joint angle (r = −0.57, *p* = 0.005), the mean thoracic flexion angle in the sagittal plane (r = −0.42, *p* = 0.049), and foot angle with respect to the floor in the sagittal plane (r = −0.59, *p* = 0.004) showed a significant negative correlation with age.

**Table 5.** Spearman's rank correlation analysis for age and floor surface-related changes in gait variables.

| | r | p |
|---|---|---|
| **Spatiotemporal parameters** | | |
| Walking speed | 0.09 | 0.683 |
| Step length on the left side | −0.35 | 0.114 |
| Step length on the right side | −0.37 | 0.093 |
| Stride length | **−0.44** | **0.042** |
| Step width | **0.60** | **0.003** |
| Cadence | **0.60** | **0.003** |
| **Kinematic variables** | | |
| Hip flexion angle at initial contact | 0.22 | 0.332 |
| Maximum hip flexion angle in 1st half of the stance phase | 0.24 | 0.292 |
| Maximum hip extension angle in the stance phase | −0.17 | 0.454 |
| Hip extension angle at toe-off | −0.20 | 0.379 |
| Maximum hip flexion angle in the swing phase | 0.01 | 0.970 |
| Peak-to-peak value of hip joint angle | −0.15 | 0.519 |
| Knee flexion angle at initial contact | −0.32 | 0.148 |
| Maximum knee flexion angle in 1st half of the stance phase | −0.15 | 0.497 |
| Maximum knee extension angle in 2nd half of stance phase | −0.21 | 0.346 |
| Knee flexion angle at toe-off | −0.07 | 0.741 |
| Maximum knee flexion angle in the swing phase | −0.12 | 0.607 |
| Peak-to-peak value of knee joint angle | −0.18 | 0.413 |
| Ankle dorsiflexion angle at initial contact | **0.59** | **0.004** |
| Maximum ankle plantarflexion angle in 1st half of stance phase | 0.28 | 0.212 |
| Maximum ankle dorsiflexion angle in 2nd half of stance phase | 0.27 | 0.232 |
| Ankle plantarflexion angle at toe-off | −0.35 | 0.105 |
| Maximum ankle dorsiflexion angle in the swing phase | **0.61** | **0.002** |
| Peak-to-peak value of ankle joint angle | **−0.57** | **0.005** |
| Foot angle at initial contact | **−0.59** | **0.004** |
| Mean value of the thoracic angle in the sagittal plane | **−0.42** | **0.049** |
| Peak-to-peak value of the thoracic angle in the sagittal plane | 0.30 | 0.172 |
| Mean value of the pelvic angle in the sagittal plane | −0.28 | 0.211 |
| Peak-to-peak value of the pelvic angle in the sagittal plane | 0.22 | 0.328 |

Bold r and *p*-values indicate a significant correlation ($p < 0.05$).

## 4. Discussion

This study examined the effects of aging on the kinematic characteristics of gait when walking on a compliant surface in healthy subjects. We found that hip and knee flexion angles during the early stance and swing phases and the ankle plantarflexion angle at IC are increased when walking on a compliant surface. Furthermore, walking on a compliant surface leads to an increase in the mean thoracic flexion angle throughout the gait cycle. The degree of increase in stride length and peak-to-peak values of the ankle joint angles when walking on the compliant surface decreased with age, while the amount of increase in the step width, cadence, ankle plantarflexion angle at IC, and in the swing phase and thoracic flexion angle throughout the gait cycle increased with age. To the best of our knowledge, this is the first study to objectively quantify the age-related changes in the trunk and lower limb kinematics when walking on a compliant surface. These changes are essential to adapting to different walking surfaces.

The increase in knee and hip flexion angles in the early stance and swing phases of gait on the compliant surface was not related to age, which was contrary to our hypothesis. Previous studies have also reported similar findings related to floor surface changes in healthy young adults [9,10]. This is a common adaptation strategy used in both young and older adults to traverse compliant surfaces by lowering the position of the CoM and increasing gait stability.

Likewise, the ankle plantarflexion angle at IC and the swing phase increases while walking on a compliant surface and this increase is amplified with advancing age. This

means that older adults tend to keep their foot in a horizontal position with respect to the floor during IC to increase contact surface area and body stability. A similar increase in the ankle plantarflexion angle at IC has been reported when walking on hard, uneven terrain, such as uneven bricks and destabilizing loose rock, in both healthy young and older adults [12,22]. In addition, the variability in the ankle plantarflexion angle at IC is greater in healthy young adults on hard, uneven terrain than on level surfaces [23]. Although previous studies were conducted on slick surfaces, not on uneven terrain, older adults showed 30% greater ankle muscle cocontraction in the early stance phase than young adults [24]. Furthermore, a study indicated that older adults tend to adjust the ankle joint angle toward plantarflexion at IC while walking on a slippery floor to reduce the required coefficient of friction (a measure of slipperiness) [25]. This adjustment of ankle joint movement is important for adapting to uneven terrain. On compliant surfaces where healthy young adults do not require adjustment of ankle joint movement, older adults, as observed in our study, tended to maintain balance by keeping their foot in a horizontal position with respect to the floor at IC instead of dorsiflexion. In other words, older adults straightaway moved to a foot-flat after IC during walking on compliant surfaces.

In addition to changes in ankle joint movement, older adults showed a greater increase in the mean thoracic flexion angles during walking on the compliant surface. Thoracic flexion plays an important role in moving the CoM of the upper body forward. Older adults reportedly employ the same movement strategy in walking on a compliant surface as during the deep-squat movement [26]. To avoid falling backward, they move their CoM forward by increasing the thoracic flexion angle during the deep-squat movement. Furthermore, the stability limits of whole-body CoM control in the anterior–posterior direction during upright standing in older adults were significantly lower than in young adults [27,28]. In addition, previous studies have shown that older adults have a significantly smaller backward shift of the center of pressure during gait initiation as compared to young adults [29,30]. Based on these findings and our results, it can be reasonably deduced that older adults prevent backward perturbations and imbalance by keeping their CoM forward and generating more thoracic flexion while walking on compliant surfaces. Strength training on trunk extensor muscles may be important for older adults to keep the trunk flexion position when walking on the compliant surface. In addition, when walking on the compliant surfaces, older adults employed a strategy of widening the base of support in the mediolateral direction by increasing the step width to prevent instability.

Regarding other spatiotemporal gait parameters, the effect of aging on the adaptive changes related to the floor surface was variable. There were no significant differences in walking speed between the level and compliant surface conditions. Interestingly, Panebianco et al. [31] reported that the walking speed on the sand was faster than that on the level surface, while Svenningsen, de Zee, and Oliveira [9] reported no significant difference in walking speed when walking on sand or level surfaces. Our findings support the latter [9]. The present results also showed that the control strategy for walking speed differs with aging. The results of our correlation analysis showed that older adults used increased cadence during gait on the compliant surface, while young adults increased their stride length. Increasing the stride length requires a transfer of the CoM over a longer distance in the single-stance phase, which is a time of high instability. Therefore, when walking on a compliant surface, older adults may prefer to increase their cadence and avoid increased instability during the single-stance phase to maintain walking speed. To improve the ability to walk on compliant surfaces, a one-leg standing exercise on the unstable surface may be useful.

Our findings revealed that when walking on the compliant surface, the amount of increase in the cadence, ankle plantarflexion angle at IC, and in the swing phase and thoracic flexion angle increased with age. These findings might be reference data for the gait characteristics on the compliant surface in healthy young and older subjects without pathological gaits. Biomechanical studies about gait on uneven terrain (e.g., compliant, rock, sand, grass, ballast, etc.) have been frequently conducted in patients with below-knee

amputation [32,33], Parkinson's disease [34–37], and children with cerebral palsy [38–40]; notably, there are only a few studies on gait characteristics of patients with hemiparesis [41]. In particular, nonarticulated ankle–foot orthoses are often used on the paretic side in patients with hemiparesis [42], which may not be able to adjust the ankle joint angle at IC when walking on the compliant surface observed in the healthy young and old adults in this study. Based on the findings of this study, future studies should investigate whether the gait kinematic characteristics on the compliant surface for patients with hemiparesis differ from those of healthy older adults, which could facilitate the development of gait rehabilitation.

There are several limitations to this study. First, we included healthy subjects with no limitations to ambulation in the living space. It is unclear how subjects with small living spaces and requiring assistance in their home develop walking adaptations. Second, only six participants over the age of 60 years were included in the study. Considering the effect of outliers on the results of interest, we visualized the results using the boxplot (Figure 4) and scatter plot (Figure 5). Although we assumed that there were no outliers that would skew the results, age-related changes may be emphasized by a bias of subjects toward the youngest and oldest categories. When conducting a two-way ANOVA with age group (young and old) as the within-subject factor and floor surface (level and compliant) as the between-subjects factor with a medium effect size (partial eta squared = 0.06), 17 subjects in each group are required. Therefore, it is necessary to investigate with larger samples of young and older adults. Third, this study compared gait data only on compliant and level surfaces. This task may presumably be easier than some of the uneven terrain walking tasks used in previous studies. Future studies should examine gait variables on other walking surfaces with different tasks (e.g., rock, sand, grass, ballast, etc.) used in previous studies.

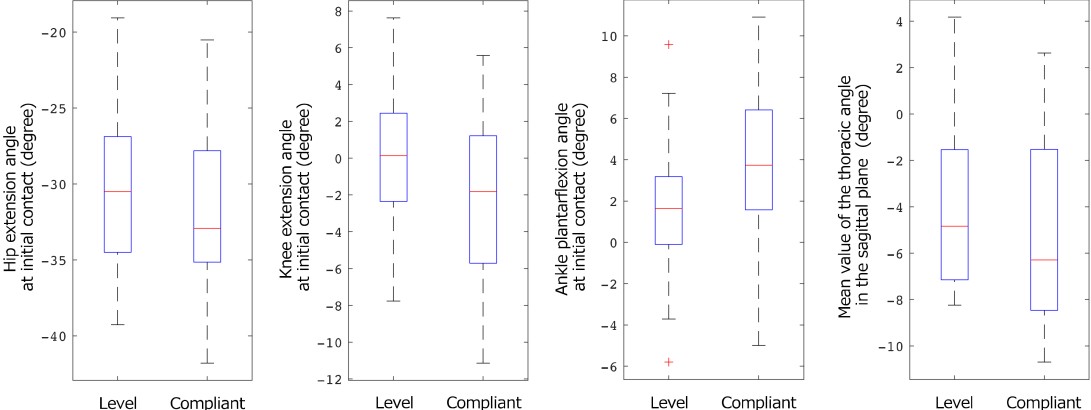

**Figure 4.** Boxplot of joint angle during gait on the level surface and compliant surface. A red plus sign is an outlier.

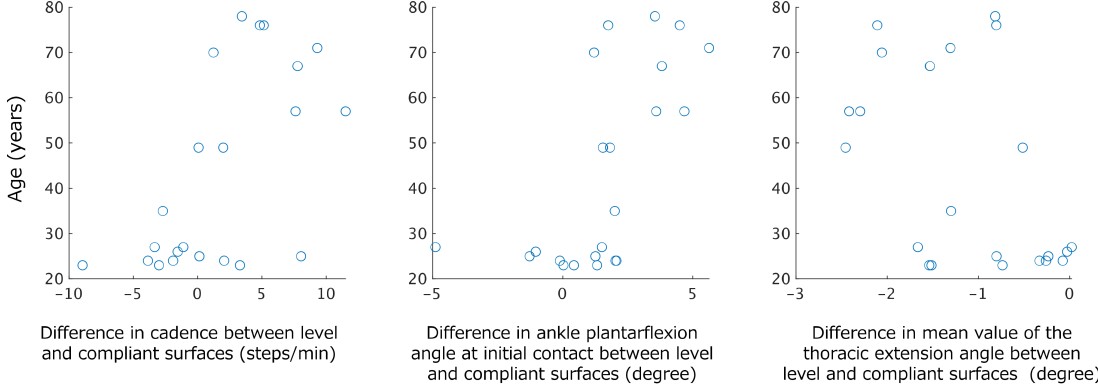

**Figure 5.** Scatter plot between age and floor surface-related changes in gait variables.

## 5. Conclusions

We found that knee flexion, hip flexion, and ankle plantarflexion angles in the early stance phase and thoracic flexion angle throughout the gait cycle are significantly increased when walking on a compliant surface compared to a level surface. The amount of increase in thoracic flexion and the ankle plantarflexion angle during walking on the compliant surface increased with aging. Older adults use these adaptation strategies along with an increased cadence to traverse a compliant surface without inducing significant instability.

**Author Contributions:** Conceptualization: K.H., Y.S. and S.-I.I.; methodology: K.H. and Y.S.; software: K.H. and Y.S.; validation: K.H. and Y.S.; formal analysis: K.H.; investigation: K.H. and Y.S.; resources: K.H., Y.S. and S.-I.I.; data curation: K.H.; writing—original draft preparation: K.H.; writing—review and editing: K.H. and Y.S. and S.-I.I.; visualization: K.H.; supervision: S.-I.I.; project administration: S.-I.I.; funding acquisition: K.H., Y.S. and S.-I.I. All authors have read and agreed to the published version of the manuscript.

**Funding:** This research was funded by JPJS KAKENHI Grant-in-Aid for Early-Career Scientists (grant number: 20K19407).

**Institutional Review Board Statement:** The study was conducted in accordance with the Declaration of Helsinki and approved by the Ethics Committee of the Tohoku University Graduate School of Medicine (protocol code: 2019-1-157, date of approval: 17 December 2018).

**Informed Consent Statement:** Informed consent was obtained from all subjects involved in the study. Written informed consent has been obtained from the participants to publish this paper.

**Data Availability Statement:** Data are available on request due to privacy and ethical restrictions.

**Acknowledgments:** We would like to thank the staff at the Department of Physical Medicine and Rehabilitation at Tohoku University for their advice and help.

**Conflicts of Interest:** The authors declare no conflict of interest.

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
