# Peer review of "Effect of Aging on the Trunk and Lower Limb Kinematics during Gait on a Compliant Surface in Healthy Individuals"

_2673-7078, doi:10.3390/biomechanics3010010_

Round 1

Reviewer 1 Report

The authors have presented a nice paper investigating the effect of aging on trunk and lower limb kinematics during gait on a compliant surface.

The paper is well written and generally clear to understand. The discussion is detailed and gives a good summary of the how the results might contribute to our understanding of mobility on older persons.

There are some minor aspects of the paper that can be improved prior to publication.

Methods

There is no indication of how many gait cycles were recorded during each trial. If one was recorded for each trial, please state this.

Line 70: A more complete description of what constitutes an abnormal respiratory status would be useful.

Lines 112-113: Were all parameters found to be non-normally distributed? If not which ones?

Results

Graphs showing the angles of each joint over the gait cycle observed for level and compliant surfaces would help the reader to understand the differences.

Discussion

The authors state that older adults tend to keep their foot in a horizontal position. This can be inferred from the ankle angles, but it would be useful to report the actual angle between the foot and the floor to support this statement.

The discussion of limitations should also mention that half the subjects were in the youngest category with few subjects in the intermediate age groups. The bias of subjects towards the youngest and oldest categories may have contributed to the age-related changes.

Reviewer 2 Report

the paper is very interesting with proper methodology 

The authors need to elaborate and put more details in the discussion about the clinical significance and possible clinical applications of the findings 

Reviewer 3 Report

Thank you for the invitation to review this manuscript entitled ‘Effect of aging on the trunk and lower limb kinematics during gait on a compliant surface in healthy individuals’. This study evaluated the gait kinematics differences of walking of hard and compliant surfaces and the correlation between age and the differences in gait kinematics.

The results showed that walking on a compliant surface increases knee and hip flexion and ankle plantarflexion, and thoracic flexion angle. The differences in  thoracic flexion angle, ankle plantarflexion angle, and cadence were significantly correlated with age. The results indicate that older adults and young adults may utilise different strategies to maintain balance when walking on a compliant surface.

The manuscript fits the scope of the special issue. In general, it is easy to follow. My major concern is that the authors seem to overlook the literature that has explored the effects of ageing on gait biomechanics, for example. Barbara et al. (reference #11) recruited older adults in their studies and explored their gait characteristics. Thus the statement in Line 49 -50 ‘However, these findings are based on healthy  young adults.’ is incorrect. The author would like to rephrase the research gap statement accordingly.

Below are other comments for consideration.

1.         More information regarding the mechanical property of the AIREX mat is needed.

2.         How many gait cycles were used to calculate the mean kinematics and gait variables?

3.         A lowpass filter with cutoff frequecy of 80Hz was used to filter the GRF data. It seems too high for me. Can the author explain or provide a reference to justify the use of 80Hz as the filtering frequency?

4.         The author provides two references for the marker set they used in the study. However, one article is written in Japanese, and the other is a book chapter where various marker set have been described. Please provide a figure to illustrate the marker placement in this study.

5.         The sample of this study is very small, a few outliers could have a huge effect on the statistics. Can the authors provide box plots for the results of interest (ie knee,hip flexion, ankle plantarflexion angles in the early stance phase and thoracic flexion angle) for the two condition comparisons? And scatter plots for the correlation analysis( thoracic flexion angle, ankle plantarflexion angle, and cadence) ?

6.         Can the authors discuss the effect of surface on step width as well?

7.         The significance of the study, as stated in Line 55 -56 is ‘….may help identify the gait parameters that need to be addressed for older adults to navigate different environments.’ Please suggest how the results inform clinical practice in gait training for an older elder.

8.         Line 245-247 The whole study studies the effect of compliance surface on gait parameters. The future study suggested by the author is about studying gait on an uneven surface in people with stroke. It seems incoherent to me.

9.         Please propose a sample size for the ‘larger samples of older adults to be studied in the future. (Line 254)

Round 2

Reviewer 2 Report

the paper is suitable for publication 

Reviewer 3 Report

The authors addressed all my comments and I am satisfied with the quality of the revised manuscript.